# Adenosine A$_{2A}$ receptor agonist polydeoxyribonucleotide ameliorates short-term memory impairment by suppressing cerebral ischemia-induced inflammation via MAPK pathway

**Il-Gyu Ko**[1], **Jun-Jang Jin**[1], **Lakkyong Hwang**[1], **Sang-Hoon Kim**[1], **Chang-Ju Kim**[1], **Jung Won Jeon**[2], **Jun-Young Chung**[3], **Jin Hee Han**[4]*

1 Department of Physiology, College of Medicine, Kyung Hee University, Seoul, Korea, 2 Department of Internal Medicine, Kyung Hee University Hospital at Gangdong, College of Medicine, Kyung Hee University, Seoul, Korea, 3 Department of Anesthesiology and Pain Medicine, Kyung Hee University Hospital at Gangdong, College of Medicine, Kyung Hee University, Seoul, Korea, 4 Department of Anesthesiology and Pain Medicine, Kyung Hee Medical Center, College of Medicine, Kyung Hee University, Seoul, Korea

* esthesi@khu.ac.kr

**Data Availability Statement:** All relevant data are within the paper and its Supporting Information files.

## Abstract

Cerebral ischemia causes tissue death owing to occlusion of the cerebral blood vessels, and cerebral ischemia activates mitogen-activated protein kinase (MAPK) and induces secretion of pro-inflammatory cytokines. Adenosine A$_{2A}$ receptor agonist, polydeoxyribonucleotide (PDRN), suppresses the secretion of pro-inflammatory cytokines and exhibits anti-inflammatory effect. In the current study, the therapeutic effect of PDRN on cerebral ischemia was evaluated using gerbils. For the induction of cerebral ischemia, the common carotid arteries were exposed, and then aneurysm clips were used to occlude the common carotid arteries bilaterally for 7 minutes. In the PDRN-treated groups, the gerbils were injected intraperitoneally with 0.3 mL of saline containing 8 mg/kg PDRN, per a day for 7 days following cerebral ischemia induction. In order to confirm the participation of the adenosine A$_{2A}$ receptor in the effects mediated by PDRN, 8 mg/kg 7-dimethyl-1-propargylxanthine (DMPX), adenosine A$_{2A}$ receptor antagonist, was treated with PDRN. In the current study, induction of ischemia enhanced the levels of pro-inflammatory cytokines and increased phosphorylation of MAPK signaling factors in the hippocampus and basolateral amygdala. However, treatment with PDRN ameliorated short-term memory impairment by suppressing the production of pro-inflammatory cytokines and inactivation of MAPK signaling factors in cerebral ischemia. Furthermore, PDRN treatment enhanced the concentration of cyclic adenosine-3,5'-monophosphate (cAMP) as well as phosphorylation of cAMP response element-binding protein (p-CREB). Co-treatment of DMPX and PDRN attenuated the therapeutic effect of PDRN on cerebral ischemia. Based on these findings, PDRN may be developed as the primary treatment in cerebral ischemia.

**Funding:** • Jin Hee Han • NRF-2017R1D1A1B03032827 • National Research Foundation of Korea • https://www.nrf.re.kr/ • Research fund support.

**Competing interests:** The authors have declared that no competing interests exist.

## 1. Introduction

Cerebral ischemia, also known as stroke, is an acute disease induced by insufficient blood supply to the brain, which can result in severe complications and a high mortality rate [1, 2]. Currently, there is no effective drug therapy for acute ischemic stroke other than intravenous or intraarterial thrombolysis. However, as the time scope for treatment with a thrombolytic agent is narrow, the utility of thrombolytic agents is very limited [3, 4].

After an ischemic stroke, the development of lesions is generally associated with severity of inflammatory reaction [5]. During stroke, inflammation-inducing mediators are created, and inflammation serves to exacerbate the symptoms and progress ischemic stroke [6]. Inflammation exacerbates ischemic damage through the secretion of pro-inflammatory cytokines such as tumor necrosis factor-$\alpha$ (TNF-$\alpha$), interleukin (IL)-1$\beta$ and IL-6 [7].

Cerebral ischemia activates mitogen-activated protein kinase (MAPK), and MAPK activation plays a selective role in determining neuronal survival or death [8]. Activation of MAPK functions primarily as a mediator for cell survival or apoptosis through phosphorylation of intracellular enzymes, transcription factors and cytoplasmic proteins [9, 10]. Extracellular signal-regulated kinase (ERK), c-Jun NH$_2$-terminal kinase (JNK) and p38 kinase are included in the MAPK system, and the MAPK signaling pathway regulates expressions of inflammatory cytokines and apoptosis factors during stroke. Therefore, this MAPK signaling pathway can be a therapeutic target when developing appropriate therapeutic agents [11, 12]. Thus, studies aimed at regulating inflammation through the MAPK system in ischemic brain tissue may provide a basis for discovering new therapeutic agents targeting stroke patients.

Adenosine and its receptors are essential neuromodulators playing important roles in various pathophysiological conditions. Adenosine A$_1$, A$_{2A}$, A$_{2B}$ and A$_{3A}$ are included in the adenosine receptors, and the adenosine receptors are expressed in inflammatory cells and immune cells. Among them, stimulation of A$_{2A}$ has been reported to reduce the secretion of pro-inflammatory cytokines in various neurological disease conditions [13, 14]. The physiological role of adenosine A$_{2A}$ receptor-induced BDNF production was demonstrated through synapse formation from immature and mature neurons, as well as protecting neurons from excitatory toxicity and increasing neurite expansion [15].

The adenosine A$_{2A}$ receptor agonist, polydeoxyribonucleotide (PDRN), exhibits an anti-inflammatory effect by inhibiting pro-inflammatory cytokine production. PDRN is also known to inhibit apoptosis in several disease states such as gastric ulcers, acute lung injury, and osteoarthritis [16–18]. Despite the excellent pharmacological effects of PDRN, few studies have evaluated the efficacy of PDRN in cerebral ischemia.

In the current study, the effect of PDRN on short-term memory and inflammation in the hippocampus after induction of transient global ischemia was evaluated using gerbils. For the experiment, the step-down avoidance task was conducted for short-term memory, and the concentrations of TNF-$\alpha$, IL-1$\beta$ and cyclic adenosine-3',5'-monophosphate (cAMP) were analyzed by enzyme-linked immunoassay (ELISA). In addition, the expression of neuronal nuclei (NeuN) was determined by immunohistochemical analysis, and the levels of adenosine A$_{2A}$ receptor, TNF-$\alpha$, IL-1$\beta$, ERK, JNK, p38, cAMP response element-binding protein (CREB) and protein kinases A (PKA) were analyzed by western blotting.

## 2. Materials and methods

### 2.1. Animals and classification

The male adult Mongolian gerbils, weighing $50 \pm 2$ g (15 weeks old), were bred in the controlled temperature ($23 \pm 2°C$) and lighting (08:00 to 20:00 h) conditions, and food and water

were provided freely. The gerbils were randomly classified into the four groups such as sham-operation group, cerebral ischemia-induced group, cerebral ischemia-induced and PDRN-treated group, cerebral ischemia-induced and PDRN with 7-dimethyl-1-propargylxanthine (DMPX)-treated group (n = 10 in each group).

This experimental procedure was approved by the Institutional Animal Care and Use Committee of Kyung Hee University and received the following approval number (KHUASP[SE]-17-071). The experimental procedures were conducted in good faith in accordance with the guidelines for animal care from the National Institutes of Health and the Korean Institute of Medical Sciences. The gerbils underwent sufficient anesthesia with Zoletil 50® (Vibac Laboratories, Carros, France) to performed surgery or sacrifice. No gerbils died or were euthanized before the end of the experiment.

## 2.2. Transient global ischemia induction

Transient global ischemia was surgically made in the same manner as described above [19, 20]. A bilateral neck incision was made after anesthetizing the gerbils by Zoletil 50® (10 mg/kg; Vibac Laboratories). Then two common carotid arteries were exposed and closed using surgical clips for 7 minutes. By a Homeothermic Blanket Control Unit (Harvard Apparatus, Massachusetts, MA, USA) that wrapped around the head and body, the temperature of the head and body was maintained at $36 \pm 0.5°C$ during the course of operation. After 7 minutes of closing, the surgical clips were removed and cerebral blood flow was allowed to reflow. Local brain blood flow on either side of the forebrain was determined using a BLF21D laser Doppler flowmeter (Transonic Systems Inc., New York, NY, USA). To prevent hypothermia, the gerbils were observed for an additional 4 hours after recovery. The animals from the sham-operation group were managed in a similar manner, but both common carotid arteries were not closed during the neck operation.

## 2.3. Treatment

The gerbils from the PDRN-treated groups were injected intraperitoneally with 0.3 mL of normal saline (0.9%) containing 8 mg/kg of PDRN (Rejuvenex®, PharmaResearch Products, Pangyo, Korea), one time per a day continued for 7 days, started a day after operation. Based on preliminary data and previous studies, the PDRN concentration considered to be the most effective was used in this experiment [17, 21]. Additionally, in order to confirm that the adenosine $A_{2A}$ receptor is involved in the effect mediated by PDRN, 8 mg/kg DMPX (Sigma Chemical Co., St. Louis, MO, USA), an adenosine $A_{2A}$ receptor antagonist, was simultaneously administered with PDRN. In the sham-operation group and in the cerebral ischemia-induced group, the gerbils received 0.3 mL of normal saline without drugs according to the same timetable. The experiment schedule is shown in Fig 1.

## 2.4. Step-down avoidance task

In the same manner as described above [22, 23], the step-down avoidance task was performed to determine the short-term memory. Eight days after induction of cerebral ischemia, the gerbils implemented a step-down avoidance task. Each gerbil was placed on a $7 \times 25$ cm platform at height of 2.5 cm, and the platform faced a grid of parallel steel bars ($45 \times 25$ cm) with a diameter of 0.1 cm and a spacing of 1 cm. Going down the grid during training, the animal immediately took the out of the box after receiving a 0.3 mA foot shock for 2 seconds. After 2 hours of training period, the latency of each gerbil was determined. During the test time, the gerbils were placed back on the platform, and the latency time was defined as the time until

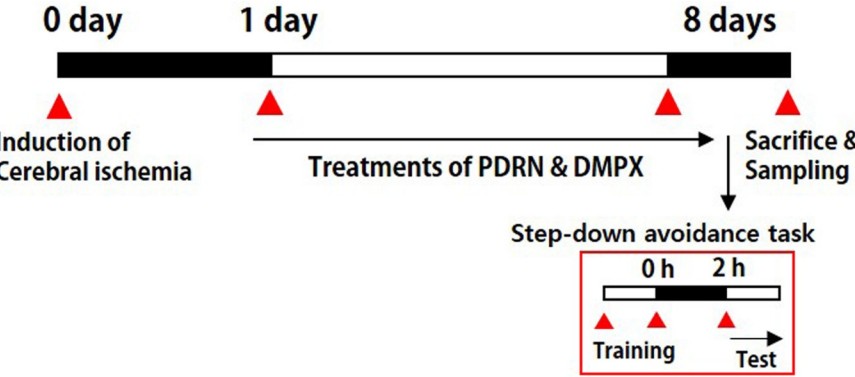

**Fig 1. Experimental schedule.**

the animal descended and placed all four feet on the grid. A delay of latency time more than 180 seconds was also calculated as 180 seconds.

## 2.5. Tissues preparation

On the 8th day after ischemia induction, immediately after measuring the latency of the step-down avoidance task, the animals were sacrificed. After anesthetizing the gerbils using the Zoletil 50® (10 mg/kg, ip; Vibac Laboratories), heart puncture was used to collect blood, left at room temperature for 1 hour and centrifugation was performed at 3,000 rpm for 20 minutes to obtain serum. After blood sampling, 50 mM phosphate-buffered saline (PBS) was transcardially perfused, and then fixed with a solution consisting of 4% paraformaldehyde in 100 mM phosphate buffer (PB, pH 7.4). After removing the brains, the brains were with the same fixative solution and then transferred to 30% sucrose solution to prevent freezing. Thereafter, 40-μm thick coronal sections were made using a freezing microtome (Leica, Nussloch, Germany), and an average of 10 sections was obtained in the CA1 region of each gerbil.

## 2.6. Pro-inflammatory cytokines and cAMP concentrations

Serum and hippocampus levels of pro-inflammatory cytokines (TNF-α and IL-1β) and cAMP concentrations were determined using enzyme-linked immunosorbent assay (ELISA). Enzyme immunoassay kits were used to detect the concentrations of TNF-α, IL-1β and cAMP in accordance with the manufacturer's instructions (Abcam, Cambridge, UK), in the same manner as described above (n = 3 in each group) [17].

## 2.7. Western blot analysis

According to the same manner as described above [24, 25], analysis of western blotting was conducted (n = 4 in each group). Priority, approximately 30 mg of hippocampal tissues were extracted using 100 mg/mL of lysis buffer. The tissues were homogenized using a lysis buffer consisting of 1 mM PMSF, 1 mM EGTA, 1 mM Na2VO$_4$, 1.5 mM MgCl$_2$·6H$_2$O, 50 mM Tris-HCl (pH 8.0), 100 mM NaF, 150 mM NaCl, 1% Triton X-100 and 10% glycerol, and then this homogeneous mixture was centrifuged at 14,000 rpm for 30 minutes. Concentration of protein was detected by a colorimetric protein analysis kit (Bio-Rad, Hercules, CA, USA). Protein 30 μg was separated from each sample on SDS-polyacrylamide gel and transferred to a nitrocellulose membrane. The antibodies for rabbit CREB antibody (1:1,000; Santa Cruz Biotechnology, Santa Cruz, CA, USA), phosphorylated (p)-CREB antibody (1:1,000; Santa Cruz

Biotechnology), rabbit PKA antibody (1:1,000; Santa Cruz Biotechnology), p-PKA antibody (1:1,000; Santa Cruz Biotechnology), rabbit ERK antibody (1:2,000; Cell Signaling Technology, Danvers, USA), rabbit p-ERK antibody (1:2,000; Cell Signaling Technology), rabbit JNK antibody (1:2,000; Cell Signaling Technology), rabbit p-JNK antibody (1:2,000; Cell Signaling Technology), rabbit p38 antibody (1:2,000; Cell Signaling Technology), rabbit p-p38 antibody, (1:2,000; Cell Signaling Technology), mouse TNF-α antibody (1:1,000; Santa Cruz Biotechnology), mouse IL-1β antibody (1:1,000; Santa Cruz Biotechnology), mouse adenosine $A_{2A}$ receptor antibody (1:1,000; Santa Cruz Biotechnology), and β-actin antibody (1:1,000; Santa Cruz Biotechnology) used were as the primary antibodies. Horseradish peroxidase-conjugated anti-mouse antibodies (1:2,000; Vector Laboratories, Burlingame, CA, USA) for β-actin, TNF-α, IL-1β, adenosine $A_{2A}$ receptor and horseradish peroxidase-conjugated anti-rabbit antibodies (1:3,000; Vector Laboratories) for p-CREP, CREB, p-PKA, PKA, p-ERK, ERK, p-JNK, JNK, p-p38, p38 were used as the secondary antibodies.

The entire experimental step was carried out under normal laboratory conditions, except for membrane transfer performed at 4˚C using pre-cooled buffer with cold pack. An enhanced chemiluminescence (ECL) detection kit (Santa Cruz Biotechnology) was used for band measurements. Each sample was loaded twice, and the number of samples was 4 per group.

## 2.8. Immunohistochemistry for NeuN

Immunohistochemistry for NeuN was conducted in the same manner as described above [26, 27]. The sections were treated with mouse anti-NeuN antibody (1:500; Abcam, Cambridge, MA, USA) overnight, and then treated with biotinylated mouse secondary antibody (1:200; Vector Laboratories, Burlingame, CA, USA) for an additional 1 hour. Secondary antibody was amplified using the Vector Elite ABC kit® (1:100; Vector Laboratories), and then 0.03% 3,3′-diaminobenzidine was used to show antibody-biotin-avidin-peroxidase complexes. The sections were then mounted on gelatin-coated slides, air-dried at room temperature overnight, and finally Permount® (Fisher Scientific, New Jersey, NJ, USA) was used to mount the coverslips.

## 2.9. Data analysis

The area of the hippocampal CA1 region on each slide was observed by optical microscope (Olympus, Tokyo, Japan), and the number of NeuN-positive cells was calculated using the Image-Pro® plus computer-assisted image analysis system (Media Cybernetics Inc., Silver Spring, MD, USA). The number of NeuN-positive cells was calculated using the following equation: $N = N_v \times V_{ref}$. N is the total number of NeuN-positive cells of the CA1 region, which is counted by multiplying the NeuN-positive cell numerical density $N_v$ by the reference volume ($mm^3$) $V_{ref}$. $N_v$ is the average numerical density of NeuN-positive cells and it is calculated based on the sum of counts within the CA1 region of each section and the volume of the CA1 region contained in each section. $V_{ref}$ is calculated according to Cavalieri's method as follows [28, 29]: $V_{ref} = a \times t \times s$. In this equation, a is the average area of the CA1 cell layer, t represents the average thickness (40 µm) of the microtome section, and s indicates the total number of sections through the reference volume.

In order to compare the relative levels of protein expressions, the bands were detected densitometrically by Molecular Analyst™ version 1.4.1 (Bio-Rad, Hercules). For relative quantification, a random value of 1.00 was given to the results of the sham-operation group (western blotting). One-way ANOVA with Duncan's post-hoc test was used for data analysis, and the results are presented as mean ± standard error of the mean. For statistical analysis, the value of $P < 0.05$ was determined to be statistically significant.

## 3. Results

### 3.1. Changes of cerebral blood flow

The flow of brain blood during carotid artery occlusion and reperfusion is shown in Fig 2. Occlusion of the carotid artery reduced brain blood flow and increased brain blood flow during reperfusion.

### 3.2. Changes of short-term memory latency and NeuN-positive cell number in the CA 1 region

Fig 3A shows the results of the latency in the step-down avoidance task. Following the induction of cerebral ischemic damage, short-term memory was impaired (P < 0.05), and treatment with PDRN ameliorated this ischemia-induced memory impairment (P < 0.05). The co-administration of PDRN and DMPX failed to increase the latency observed with PDRN in cerebral ischemia.

Fig 3B and 3C are the photomicrographs of NeuN-positive cells in the hippocampal CA1 region. Cerebral ischemic damage decreased the number of NeuN-positive cells, indicating reduced neuronal survival in the hippocampal CA1 region (P < 0.05), and treatment with PDRN improved neuronal survival in cerebral ischemia (P < 0.05). The co-administration of PDRN and DMPX failed to enhance the neuronal survival observed with PDRN in cerebral ischemia.

### 3.3. Changes of pro-inflammatory cytokine expression

To determine whether PDRN improves cerebral ischemia, ELISA and western blot analysis were performed by examining the effect on PDRN on production of pro-inflammatory cytokines, TNF-α (Fig 4A) and IL-1β (Fig 4B). Following cerebral ischemic damage, TNF-α and IL-1β expressions were enhanced in the serum and hippocampus (P < 0.05), and treatment with PDRN suppressed the expressions of TNF-α and IL-1β (P < 0.05). The co-administration of PDRN and DMPX failed to decrease TNF-α and IL-1β expressions observed with PDRN in cerebral ischemia.

### 3.4. Changes of phosphorylation of MAPK cascade

To determine whether cerebral ischemia is improved by PDRN, the effect of PDRN on MAPK phosphorylation was investigated using western blotting (Fig 5). Induction of cerebral ischemia promoted MAPK cascade phosphorylation, such as ERK, JNK and p38 (P < 0.05). Interestingly, treatment with PDRN more enhanced the phosphorylation of ERK, JNK and p38

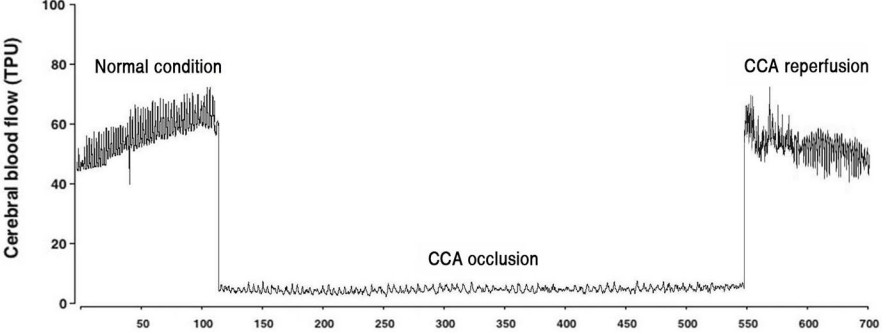

**Fig 2. Brain blood flow during occlusion and reperfusion of both common carotid arteries.**

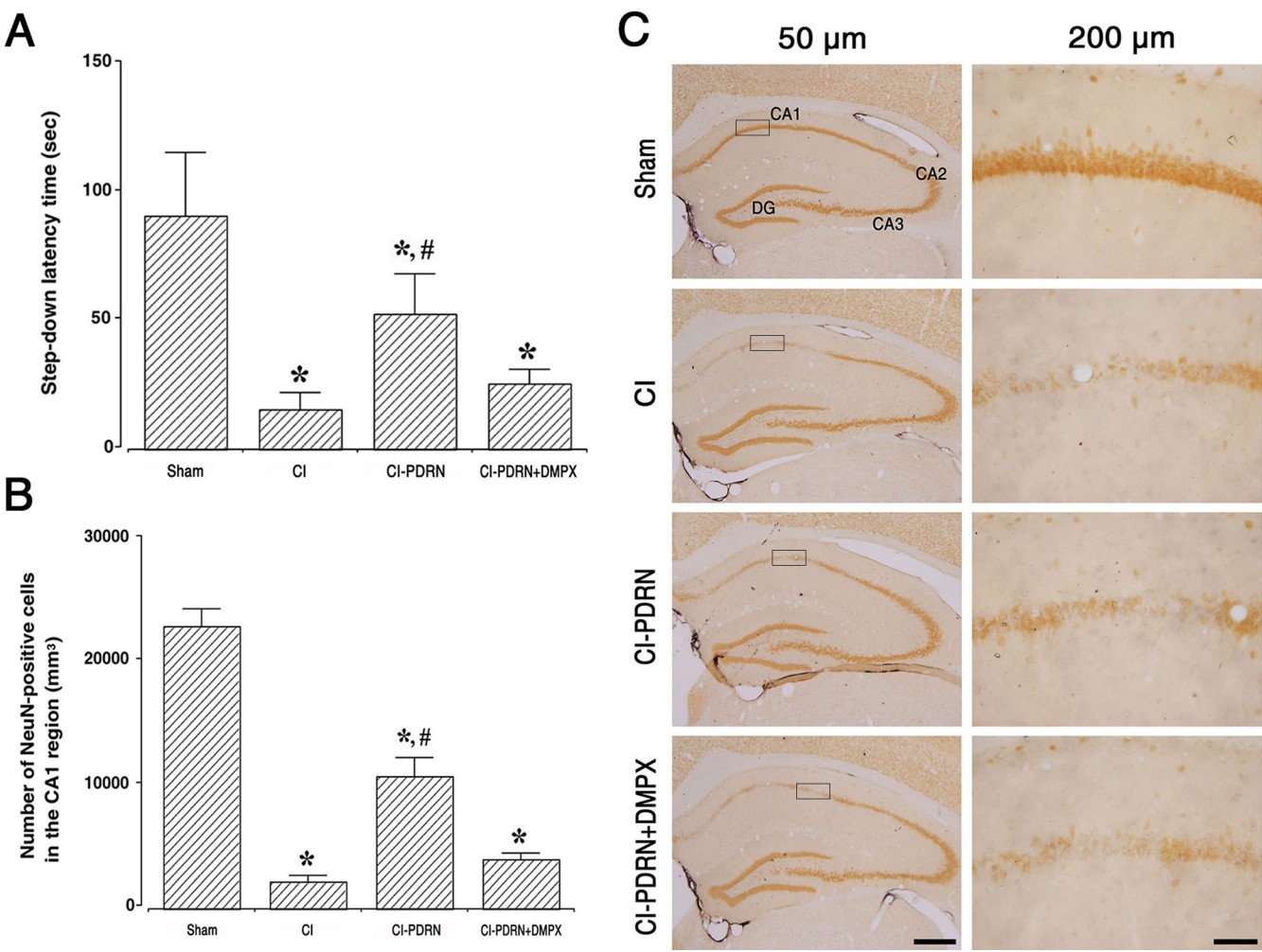

**Fig 3. Changes in short-term memory and neuronal survival in the CA1 region.** A. Latency of the step-down avoidance task in each group. B. Number of NeuN-positive cells in each group. C. Photomicrographs of NeuN-positive cells in the hippocampal CA1 region. The scale bar represents 50 μm (left) and 200 μm (right). (□) Area of magnification at 200 μm in CA1 region. Sham, sham-operation group; CI, cerebral ischemia-induced group; CI-PDRN, cerebral ischemia-induced and polydeoxyribonucleotide (PDRN)-treated group; CI-PDRN+DMPX, cerebral ischemia-induced and PDRN with 7-dimethyl-1-propargylxanthine (DMPX)-treated group. * indicates P < 0.05 compared with the sham-operation group. # indicates P < 0.05 compared with the cerebral ischemia-induced group.

(P < 0.05). The co-administration of PDRN and DMPX failed to further enhance ERK, JN and p38 phosphorylation observed with PDRN in cerebral ischemia.

## 3.5. Changes of cAMP concentration and adenosine A$_{2A}$ receptor expression

The cAMP concentration in the serum and hippocampus and adenosine A$_{2A}$ receptor expression in the hippocampus are shown in Fig 6. Induction of cerebral ischemia decreased the levels of cAMP concentration and adenosine A$_{2A}$ receptor expression (P < 0.05), and treatment with PDRN improved the levels of cAMP concentration and adenosine A$_{2A}$ receptor expression (P < 0.05). The co-administration of PDRN and DMPX failed to increase the levels of cAMP concentration and adenosine A$_{2A}$ receptor expression observed with PDRN in cerebral ischemia.

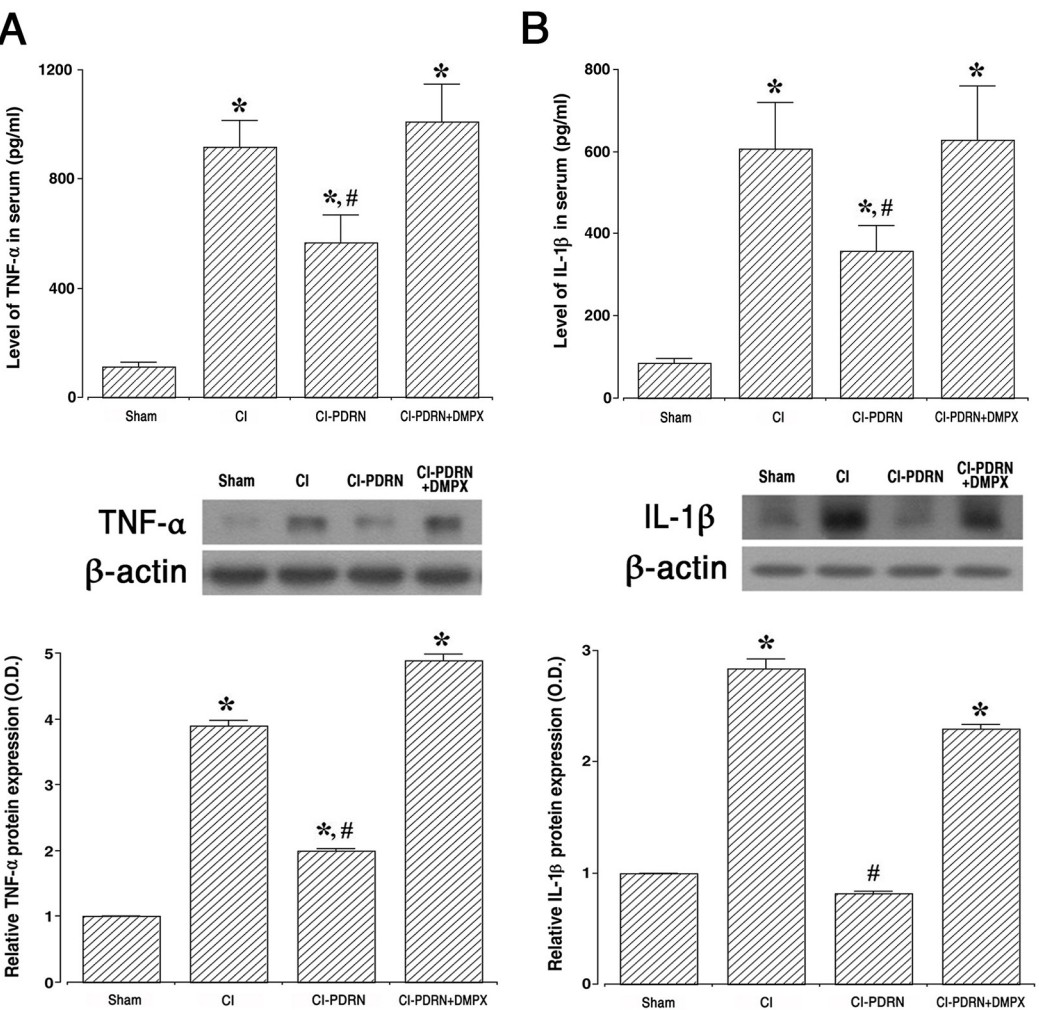

**Fig 4. Altered expression of pro-inflammatory cytokines in the serum and hippocampus.** A-upper. Concentration of tumor necrosis factor-α (TNF-α) in the serum. A-lower. The relative level of TNF-α in the hippocampus. B-upper. Concentration of interleukin (IL)-1β in the serum. B-lower. The relative level of IL-1β in the hippocampus. Sham, sham-operation group; CI, cerebral ischemia-induced group; CI-PDRN, cerebral ischemia-induced and polydeoxyribonucleotide (PDRN)-treated group; CI-PDRN+DMPX, cerebral ischemia-induced and PDRN with 7-dimethyl-1-propargylxanthine (DMPX)-treated group. * indicates P < 0.05 compared with the sham-operation group. # indicates P < 0.05 compared with the cerebral ischemia-induced group.

### 3.6. Changes of ratio in p-CREB vs CREB and p-PKA vs PKA

Western blot analysis was used to measure the relative expressions of p-CREB vs CREB and p-PKA vs PKA (Fig 7). Induction of cerebral ischemia decreased the ratio of p-CREB vs CREB and the ratio of p-PKA vs PKA when compared with the sham-operation group (P < 0.05). Treatment with PDRN increased the ratio of p-CREB vs CREB and ratio of p-PKA vs PKA in the hippocampus (P < 0.05). The co-administration of PDRN and DMPX failed to enhance the phosphorylation of CREB and PKA observed with PDRN in cerebral ischemia.

## 4. Discussion

Damage by ischemia particularly destroys pyramidal neurons in the hippocampal CA1 region, and ischemia induces apoptotic cell death in the hippocampal CA1 neurons [19, 30].

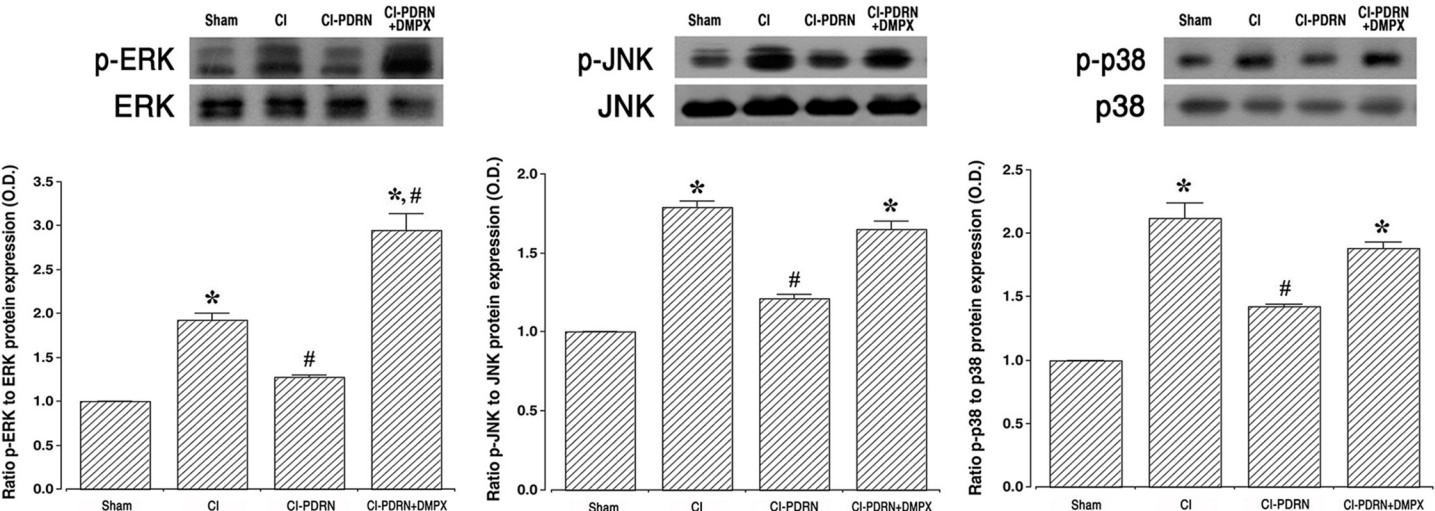

**Fig 5. Changes in the mitogen-activated protein kinase (MAPK) cascade in the hippocampus.** Left. Ratio of phosphorylated extracellular signal-regulated kinases (p-ERK) to ERK. Middle. Ratio of phosphorylated c-Jun $NH_2$-terminal kinases (p-JNK) to JNK. Right. Ratio of phosphorylated p38 kinase (p-p38) to p38 in the hippocampus. Sham, sham-operation group; CI, cerebral ischemia-induced group; CI-PDRN, cerebral ischemia-induced and polydeoxyribonucleotide (PDRN)-treated group; CI-PDRN+DMPX, cerebral ischemia-induced and PDRN with 7-dimethyl-1-propargylxanthine (DMPX)-treated group. * indicates $P < 0.05$ compared with the sham-operation group. # indicates $P < 0.05$ compared with the cerebral ischemia-induced group.

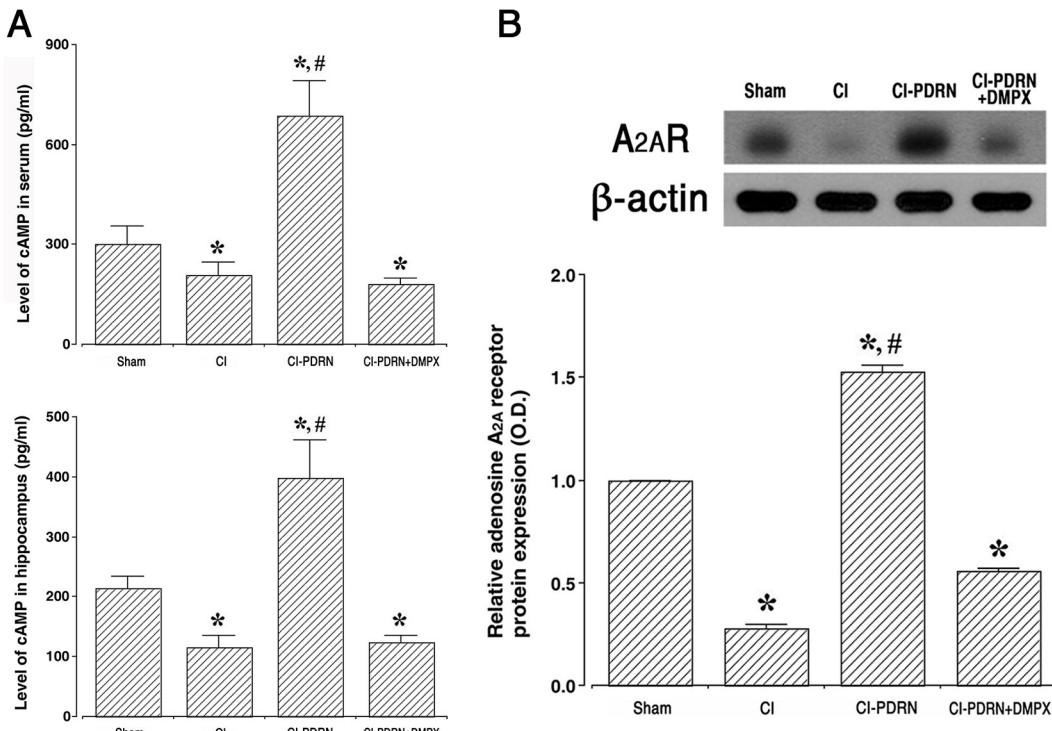

**Fig 6. Changes in cAMP concentration and adenosine $A_{2A}$ receptor expression.** A-upper. Concentration of cAMP in serum. A-lower. Concentration of cAMP in the hippocampus. B. The relative expression of the adenosine $A_{2A}$ receptor in the hippocampus. Sham, sham-operation group; CI, cerebral ischemia-induced group; CI-PDRN, cerebral ischemia-induced and polydeoxyribonucleotide (PDRN)-treated group; CI-PDRN+DMPX, cerebral ischemia-induced and PDRN with 7-dimethyl-1-propargylxanthine (DMPX)-treated group. * indicates $P < 0.05$ compared with the sham-operation group. # indicates $P < 0.05$ compared with the cerebral ischemia-induced group.

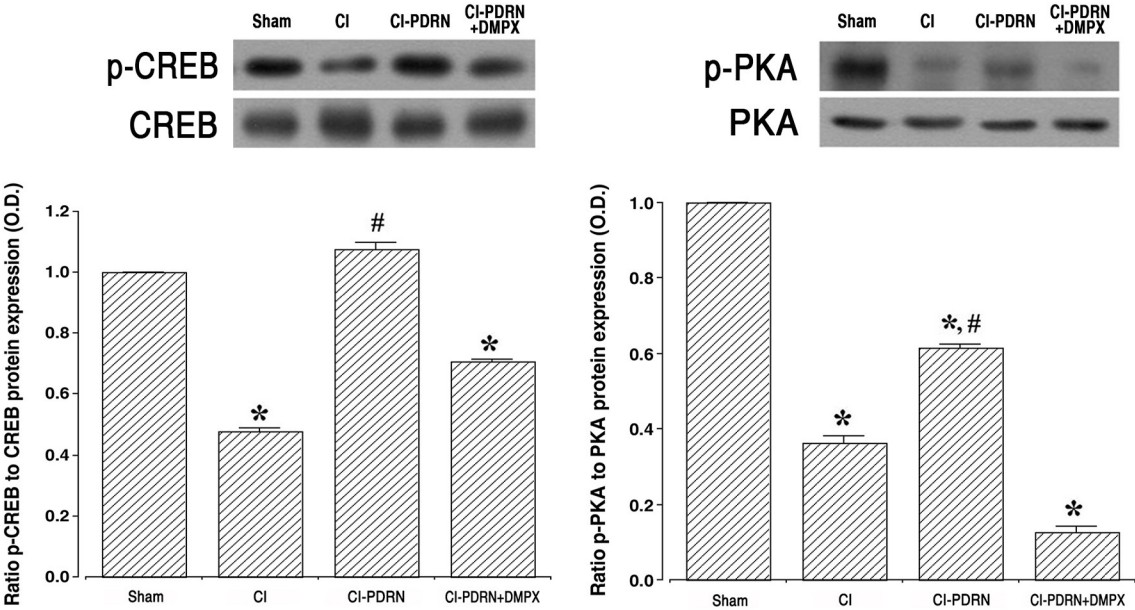

**Fig 7. Changes in phosphorylated cAMP response element-binding protein (p-CREB) to CREB ratio and phosphorylated protein kinases A (p-PKA) to PKA ratio.** Left. Ratio of p-CREB to CREB in the hippocampus. Right. Ratio of p-PKA to PKA in the hippocampus. Sham, Sham-operation group; CI, cerebral ischemia-induced group; CI-PDRN, cerebral ischemia-induced and polydeoxyribonucleotide (PDRN)-treated group; CI-PDRN+DMPX, cerebral ischemia-induced and PDRN with 7-dimethyl-1-propargylxanthine (DMPX)-treated group. * indicates P < 0.05 compared with the sham-operation group. # indicates P < 0.05 compared with the cerebral ischemia-induced groups.

Pyramidal neurons are crucial for learning and memory, and appearance of passive avoidance memory impairment after ischemia is associated with damage in the neurons of the CA1 region [31, 32]. In the current study, the cerebral ischemic injury resulted in cell loss in the hippocampal CA1 neurons, and this neuronal cell loss reduced short-term memory when compared with the gerbils of the sham-operation group. The present findings are similar to previous studies showing that loss of neurons in the CA1 region caused short-term memory impairment [31, 33].

The ischemic brain exhibits inflammation characterized by the accumulation of inflammatory cells and mediators. Previous studies have suggested that induction of ischemic damage enhances neuronal cell loss owing to increased levels of inflammatory exudates and pro-inflammatory cytokines [6, 7, 34]. Furthermore, an increment of inflammatory cytokines in the brain acts as a major causes of neuronal cell loss and memory impairment [35, 36]. Based on current findings, enhanced secretion of TNF-α and IL-1β, pro-inflammatory cytokines, in the serum, hippocampus, and basolateral amygdala (Supplement 1 in S1 File) exacerbated the symptoms of ischemic injury. These results indicate that symptoms were worsened by excessive production of pro-inflammatory cytokines during cerebral ischemia. Inhibiting the secretion of pro-inflammatory cytokines is one of the important treatment strategies for the brain ischemic injury.

Most of cells involved in wound healing express the adenosine $A_{2A}$ receptor [17], and this adenosine $A_{2A}$ receptor is locates in several brain regions and modulates the pathophysiological response to ischemic stroke [14]. Agonists on adenosine $A_{2A}$ receptor have been reported to be useful for the treating of inflammatory diseases [14, 37]. In the previous studies, adenosine $A_{2A}$ receptor knockout mice exhibited symptoms of chronic cerebral ischemia such as working memory impairment, increased demyelination, glial proliferation, and increased pro-

inflammatory cytokines [4, 38]. Reduced functional capacity of BDNF in adenosine $A_{2A}$ receptor knockout mice was associated with a decrease in hippocampal BDNF level, and pharmacological blockade of adenosine $A_{2A}$ receptors significantly reduced BDNF level in the hippocampus of normal mice [39]. These results indicated that tonic activation of adenosine $A_{2A}$ receptor is required for BDNF-induced potentiation of synaptic transmission and for sustaining a normal BDNF tone in the hippocampus [39]. The facilitating action of BDNF on hippocampal long-term potentiation is critically dependent on the presence of extracellular adenosine and activation of the $A_{2A}$ receptor through a cAMP/PKA-dependent mechanism [40]. Activation of the adenosine $A_{2A}$ receptor regulates BDNF production in rat cortical neurons, which provides neuroprotective action [15].

In the current study, PDRN treatment substantially suppressed the secretion of pro-inflammatory cytokines. Inflammation is a compensatory response to cellular and tissue damage caused by ischemia-reperfusion injury, and the inflammatory response is primarily regulated by the signaling pathway of the MAPK cascade [8, 12]. MAPK is essential for the regulation of various inflammatory mediators, and MAPK is a kind of kinases that regulate cellular response to external stress signals or inflammatory cytokines [2, 41]. It was demonstrated that the induction of ischemic damage activated phosphorylation of the MAPK cascade, and this activation of MAPK controls a wide range of cellular processes [2, 12]. Our current study found that phosphorylation of the MAPK cascade pathway was increased by ischemic damage and this phosphorylation of the MAPK cascade pathway was reduced by PDRN treatment.

The intracellular concentration of cAMP is increased by stimulation of the adenosine $A_{2A}$ receptor, and this increased cAMP concentration serves as a physiological inhibitor to function of inflammatory neutrophil [42]. Furthermore, adenosine $A_{2A}$ receptor activation promotes signal from the cAMP-PKA pathway and accelerates the level of CREB phosphorylation. Elevated cAMP concentration suppresses the level of phosphorylation of the MAPK cascade pathway in stimulated cells [42, 43]. Adenosine $A_{2A}$ receptor agonist, PDRN, has been proposed to have therapeutic potential in inflammatory diseases [16, 17]. In the current study, PDRN treatment increased the cAMP concentration in gerbils presenting cerebral ischemia, and this increased cAMP concentration inhibited phosphorylation of the MAPK cascade pathway, thereby inactivating the MAPK cascade pathway in the hippocampus and basolateral amygdala (Supplements 2–4 in S1 File).

The current study has revealed that PDRN treatment inhibits inflammation, improves neuronal cell survival, and prevents a decline in short-term memory in a brain ischemia animal model (S1 Graphic abstract). Co-administration of PDRN and adenosine $A_{2A}$ receptor antagonist DMPX attenuated the therapeutic effect of PDRN in cerebral ischemia. Based on these findings, PDRN may be developed as the primary treatment in cerebral ischemia.

## Supporting information

**S1 File.**
(DOCX)

**S1 Graphic abstract.**
(TIF)

**S1 Raw images.**
(PDF)

## Author Contributions

**Conceptualization:** Il-Gyu Ko, Jin Hee Han.

**Formal analysis:** Jun-Jang Jin, Lakkyong Hwang.

**Funding acquisition:** Jin Hee Han.

**Investigation:** Sang-Hoon Kim, Jun-Young Chung.

**Methodology:** Jun-Jang Jin, Lakkyong Hwang, Sang-Hoon Kim, Jung Won Jeon.

**Resources:** Il-Gyu Ko, Jin Hee Han.

**Supervision:** Jin Hee Han.

**Validation:** Chang-Ju Kim, Jun-Young Chung.

**Writing – original draft:** Il-Gyu Ko.

**Writing – review & editing:** Chang-Ju Kim, Jin Hee Han.

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
