## [Decision Letter · Decision Letter 0]

18 Dec 2020

PONE-D-20-25445

Adenosine A2A receptor agonist polydeoxyribonucleotide ameliorates short-term memory impairment by suppressing cerebral ischemia-induced inflammation via MAPK pathway

PLOS ONE

Dear Dr. Han,

Thank you for submitting your manuscript to PLOS ONE. After careful consideration, we feel that it has merit but does not fully meet PLOS ONE’s publication criteria as it currently stands. Therefore, we invite you to submit a revised version of the manuscript that addresses the points raised during the review process.

We look forward to receiving your revised manuscript.

Kind regards,

Giuseppe Pignataro, MD, PhD

Academic Editor

PLOS ONE

Journal Requirements:

2. As part of your revisions, we kindly request additional details pertaining to animal care, use and research procedures. Please address the following items in your Methods section: (1) all methods undertaken to minimize potential pain and distress; (2) environmental enrichments that the gerbils were provided (if applicable); (3) the number of animals that died prior to the experimental endpoints and the cause of death for any that were found dead; (4) the monitoring parameters and the humane endpoints/criteria for early euthanasia for animals who became severely ill. We thank you for your attention to this matter.

Reviewers' comments:

Reviewer's Responses to Questions

**Comments to the Author**

1. Is the manuscript technically sound, and do the data support the conclusions?

Reviewer #1: Yes

Reviewer #2: Partly

2. Has the statistical analysis been performed appropriately and rigorously? 

Reviewer #1: Yes

Reviewer #2: I Don't Know

3. Have the authors made all data underlying the findings in their manuscript fully available?

Reviewer #1: Yes

Reviewer #2: Yes

4. Is the manuscript presented in an intelligible fashion and written in standard English?

Reviewer #1: Yes

Reviewer #2: Yes

5. Review Comments to the Author

Reviewer #1: Results

In the present work Il-Gyu Ko and colleagues demonstrated that the administration of an Adenosine A2a receptor agonist, polydeoxyribonucleotide (PDRN), following the induction of transient global ischemia in gerbils results in an improvement in short-term memory tested 8 days following induction of ischemia and in the survival of NeuN-positive cells in the CA1 region of Hippocampus. PDRN induced a decrease in proinflammatory cytokine production and in the activation of MAPK signalling which were increased following ischemia. Moreover an increase in the concentration of C-AMP in serum and hippocampus and of A2a receptor expression, accompanied by an increase of CREB and PKA phosphorylation and a decrease of the phosphorylation of ERK, p38 and JNK were observed. The co-treatment with the A2a receptor antagonist DMPX attenuated the therapeutic effect of PDRN.

The manuscript provides a demonstration of the efficacy of an adenosine A2a receptor agonist on cognitive decline following cerebral ischemia by inhibiting the activation of endogenous inflammation mechanisms.

However, the neuroprotective effect of adenosine neurotransmission in ischemia is not emphasized in the introduction and a greater emphasis could underline the involvement of the pathway involved in neuronal plasticity in the discussion.

English should be checked in all the text and in some places there is no correspondence between text and figures.

Regarding the treatment, the injection volume is unclear as different values are reported in the abstract and in the text.

In Figure 2C the spatial reference of the magnification at 200 μm is missing.

Regarding the cognitive task it is not clear when the training phase was carried out with respect to induction of ischemia.

Moreover, only the hippocampus was considered in the measurements. Other brain areas, for example the basolateral amygdala, are also important in the acquisition and consolidation of the memory trace relating to the association between the shock and the position of the animal.

Reviewer #2: Paper by Jin-Hee Han and collegues describes the effects of adenosine A2A receptor agonist, polydeoxyribonucleotide (PDRN), on short-term memory in gerbils subjected to global ischemia. The authors speculate that treatment with PDRN ameliorated short-term memory impairment by suppressing the production of pro-inflammatory cytokines and inactivation of MAPK signaling factors in cerebral ischemia. The work is discreet, well written and is the first to analyze the effect of polydeoxyribonucleotide (PDRN) in an ischemia model. A couple of issues need to be resolved:

1. There is no indication of the calculation of the sample size for in vivo experiments together with the number of experimental groups for western blotting evaluations etc.

2. What evidence is available regarding the ability of the experimental compound PNDR to cross the blood brain barrier when administered intraperitoneally?

6. PLOS authors have the option to publish the peer review history of their article (what does this mean?). If published, this will include your full peer review and any attached files.

Reviewer #1: No

Reviewer #2: **Yes: **Antonio Vinciguerra

---

## [Author Response · Author response to Decision Letter 0]

5 Feb 2021

Answer to Reviewers’ Comments

Manuscript ID: PONE-D-20-25445

Title: Adenosine A2A receptor agonist polydeoxyribonucleotide ameliorates short-term memory impairment by suppressing cerebral ischemia-induced inflammation via MAPK pathway

Authors: Il-Gyu Ko, Jun-Jang Jin, Lakkyong Hwang, Sang-Hoon Kim, Chang-Ju Kim, Jung Won Jeon, Jun-Young Chung, Jin Hee Han

We sincerely appreciate for your kind advice and comments to our manuscript. We revised the manuscript according to the reviewer’s comments. We added new experimental data, and modifications were expressed in red (Please check the attached file). 

Reviewer #1: Results

In the present work Il-Gyu Ko and colleagues demonstrated that the administration of an Adenosine A2a receptor agonist, polydeoxyribonucleotide (PDRN), following the induction of transient global ischemia in gerbils results in an improvement in short-term memory tested 8 days following induction of ischemia and in the survival of NeuN-positive cells in the CA1 region of Hippocampus. PDRN induced a decrease in proinflammatory cytokine production and in the activation of MAPK signalling which were increased following ischemia. Moreover an increase in the concentration of C-AMP in serum and hippocampus and of A2a receptor expression, accompanied by an increase of CREB and PKA phosphorylation and a decrease of the phosphorylation of ERK, p38 and JNK were observed. The co-treatment with the A2a receptor antagonist DMPX attenuated the therapeutic effect of PDRN.

Q1. The manuscript provides a demonstration of the efficacy of an adenosine A2a receptor agonist on cognitive decline following cerebral ischemia by inhibiting the activation of endogenous inflammation mechanisms. However, the neuroprotective effect of adenosine neurotransmission in ischemia is not emphasized in the introduction and a greater emphasis could underline the involvement of the pathway involved in neuronal plasticity in the discussion.

Answer 1. According to reviewer comment, we have added sentences to emphasize the neuroprotective effect of adenosine A2A receptor in the manuscript.

• Following sentence was added into the Introduction.

The physiological role of adenosine A2A receptor-induced BDNF production was demonstrated through synapse formation from immature and mature neurons, as well as protecting neurons from excitatory toxicity and increasing neurite expansion [15].

• Following sentences were added into the Discussion.

Reduced functional capacity of BDNF in adenosine A2A receptor knockout mice was associated with a decrease in hippocampal BDNF level, and pharmacological blockade of adenosine A2A receptors significantly reduced BDNF level in the hippocampus of normal mice [39]. These results indicated that tonic activation of adenosine A2A receptor is required for BDNF-induced potentiation of synaptic transmission and for sustaining a normal BDNF tone in the hippocampus [39]. The facilitating action of BDNF on hippocampal long-term potentiation is critically dependent on the presence of extracellular adenosine and activation of the A2A receptor through a cAMP/PKA-dependent mechanism [40]. Activation of the adenosine A2A receptor regulates BDNF production in rat cortical neurons, which provides neuroprotective action [15].

• Following references were added into the Reference list.

15. Jeon SJ, Rhee SY, Ryu JH, Cheong JH, Kwon K, Yang SI, et al. Activation of adenosine A2A receptor up-regulates BDNF expression in rat primary cortical neurons. Neurochem Res. 2011; 36(12):2259-69. http://doi.org/10.1007/s11064-011-0550-y.

39. Tebano MT, Martire A, Potenza RL, Grò C, Pepponi R, Armida M, et al. Adenosine A2A receptors are required for normal BDNF levels and BDNF-induced potentiation of synaptic transmission in the mouse hippocampus. J Neurochem. 2008; 104(1):279-86. http://doi.org/10.1111/j.1471-4159.2007.05046.x. 

40. Fontinha BM, Diógenes MJ, Ribeiro JA, Sebastião AM. Enhancement of long-term potentiation by brain-derived neurotrophic factor requires adenosine A2A receptor activation by endogenous adenosine. Neuropharmacology. 2008; 54(6):924-33. http://doi.org/10.1016/j.neuropharm.2008.01.011. 

Q2. English should be checked in all the text and in some places there is no correspondence between text and figures.

Answer 2. According to reviewer comment, we carefully checked spelling and grammar throughout text. Also, the text and figure have been modified to match. 

• (Fig. 1) was deleted from the “Transient global ischemia induction” of the Materials and Methods.

Local brain blood flow on either side of the forebrain was determined using a BLF21D laser Doppler flowmeter (Transonic Systems Inc., New York, NY, USA) (Fig. 1).

• Fig. 1 was inserted in the “Treatment” of the Materials and Methods.

The experiment schedule is shown in Fig 1.

• Following words were modified in the Results.

To determine whether PDRN improves cerebral ischemia, ELISA and western blot analysis were performed by examining the effect on PDRN on production of pro-inflammatory cytokines, TNF-α (Fig. 4A) and IL-1β (Fig. 4B).

• Following hyphens were added in the Fig. 4 and Fig. 6.

Fig 4. Altered expression of pro-inflammatory cytokines in the serum and hippocampus. A-upper. Concentration of tumor necrosis factor-α (TNF-α) in the serum. A-lower. The relative level of TNF-α in the hippocampus. B-upper. Concentration of interleukin (IL)-1β in the serum. B-lower. The relative level of IL-1β in the hippocampus. Sham, sham-operation group; CI, cerebral ischemia-induced group; CI-PDRN, cerebral ischemia-induced and polydeoxyribonucleotide (PDRN)-treated group; CI-PDRN+DMPX, cerebral ischemia-induced and PDRN with 7-dimethyl-1-propargylxanthine (DMPX)-treated group. * indicates P < 0.05 compared with the sham-operation group. # indicates P < 0.05 compared with the cerebral ischemia-induced group.

Fig 6. Changes in cAMP concentration and adenosine A2A receptor expression. A-upper. Concentration of cAMP in serum. A-lower. Concentration of cAMP in the hippocampus. B. The relative expression of the adenosine A2A receptor in the hippocampus. Sham, sham-operation group; CI, cerebral ischemia-induced group; CI-PDRN, cerebral ischemia-induced and polydeoxyribonucleotide (PDRN)-treated group; CI-PDRN+DMPX, cerebral ischemia-induced and PDRN with 7-dimethyl-1-propargylxanthine (DMPX)-treated group. * indicates P < 0.05 compared with the sham-operation group. # indicates P < 0.05 compared with the cerebral ischemia-induced group.

Q3. Regarding the treatment, the injection volume is unclear as different values are reported in the abstract and in the text.

Answer 3. According to reviewer comment, we corrected injection volume in the Abstract.

• Following word was modified in the Abstract.

In the PDRN-treated groups, the gerbils were injected intraperitoneally with 0.3 mL of saline containing 8 mg/kg PDRN, per a day for 7 days following cerebral ischemia induction.

Q4. In Figure 2C the spatial reference of the magnification at 200 μm is missing.

Answer 4. According to reviewer comment, we have marked the reference of the enlarged figure in the form of a box.

• Following figure was modified in the Results.

Q5. Regarding the cognitive task it is not clear when the training phase was carried out with respect to induction of ischemia.

Answer 5. The step-down avoidance task evaluates short-term memory ability. In this study, training was performed 8 days after the induction of cerebral ischemia, and testing was performed 2 hours after training. This was explained in the “Step-down avoidance task” of the

Materials and Methods.

According to reviewer comment, the experiment schedule has been newly inserted to make it easier to check the experiment schedule.

• Following sentence was added in the Materials and Methods.

The experiment schedule is shown in Fig 1.

• Following figure was added in the Results.

Q6. Moreover, only the hippocampus was considered in the measurements. Other brain areas, for example the basolateral amygdala, are also important in the acquisition and consolidation of the memory trace relating to the association between the shock and the position of the animal.

Answer 6. According to reviewer comment, we further evaluated the factors identified in the hippocampal region in the basolateral amygdala region. In the isolated amygdala, the concentration of inflammatory cytokines (TNF-α and IL-1β) was evaluated using ELISA. The expression of MAPK cascade (ERK, JNK, and p38), adenosine A2A receptor, CREB, and PKA was confirmed using western blotting. The results of additional research are same as in this study, and the results are shown in supplement 1-4.

In the results of additional study, concentration of pro-inflammatory cytokines and expression of MAPK cascade were increased in the basolateral amygdala following cerebral ischemic insult. Expression of adenosine A2A receptor, p-CREB, and p-PKA was decreased in the basolateral amygdala by cerebral ischemic damage. However, PDRN treatment suppressed concentration of pro-inflammatory cytokines and expression of MAPK cascade, whereas enhanced adenosine A2A receptor, p-CREB, and p-PKA expression in the basolateral amygdala. The co-treatment of PDRN and DMPX did not show any improvement effect in basolateral amygdala, as shown in the hippocampus. Additional study results indicate that PDRN treatment is effective in improving cerebral ischemia by acting effectively on the basolateral amygdala. 

• Following sentences were added in the Discussion.

Based on current findings, enhanced secretion of TNF-α and IL-1β, pro-inflammatory cytokines, in the serum, hippocampus, and basolateral amygdala (supplement 1) exacerbated the symptoms of ischemic injury. 

In the current study, PDRN treatment increased the cAMP concentration in gerbils presenting cerebral ischemia, and this increased cAMP concentration inhibited phosphorylation of the MAPK cascade pathway, thereby inactivating the MAPK cascade pathway in the hippocampus and basolateral amygdala (supplement 2-4). 

• Following figures and figure legends were added in the supplement results.

Reviewer #2

Paper by Jin-Hee Han and collegues describes the effects of adenosine A2A receptor agonist, polydeoxyribonucleotide (PDRN), on short-term memory in gerbils subjected to global ischemia. The authors speculate that treatment with PDRN ameliorated short-term memory impairment by suppressing the production of pro-inflammatory cytokines and inactivation of MAPK signaling factors in cerebral ischemia. The work is discreet, well written and is the first to analyze the effect of polydeoxyribonucleotide (PDRN) in an ischemia model. A couple of issues need to be resolved:

Q1. There is no indication of the calculation of the sample size for in vivo experiments together with the number of experimental groups for western blotting evaluations etc.

Answer 1. According to reviewer comment, we have inserted the sentences about the sample size for in vivo experiments together with the number of experimental groups for western blotting evaluations in the manuscript.

• Following sentences were added in the “Western blot analysis” of the Materials and Methods.

According to the same manner as described above [24,25], analysis of western blotting was conducted (n = 4 in each group). Priority, approximately 30 mg of hippocampal tissues were extracted using 100 mg/mL of lysis buffer.

Each sample was loaded twice, and the number of samples was 4 per group.

Q2. What evidence is available regarding the ability of the experimental compound PNDR to cross the blood brain barrier when administered intraperitoneally?

Answer 2. When considering the pharmacological action of PDRN, there is reason to infer that it acts on the brain through the BBB. The rationale for the action of PDRN can be classified into two categories: 1. Adenosine receptor signaling and BBB permeability regulation and intracellular action. 2. Adenosine receptor signaling increases immune cell entry by increasing BBB permeability. On this basis, it is likely that an active carried-mediated transport enables PDRN to traverse the blood-brain barrier as already reported for nucleosides.

1. Adenosine receptor signaling and BBB permeability regulation and intracellular action.

Studies have shown that adenosine-based drugs may play a substantial modulatory role in CNS barrier permeability. Koszalka et al. (2004) reported that adenosine produced extracellularly during disease in an experimental autoimmune encephalomyelitis (EAE) animal model positively regulates lymphocyte entry into the brain and spinal cord. Mills et al. (2008) suggested that experimental recruitment of adenosine receptors either by the broad-spectrum agonist NECA or the engagement of both A1 and A2A receptors by selective agonists (CCPA and CGS21680) cumulatively and transiently augmented BBB permeability facilitating the entry of intravenously infused macromolecules into the CNS. Furthermore, the analysis of engineered mice lacking adenosine receptors reveals a limited entry of macromolecules into the brain upon exposure to adenosine receptor agonists. 

CNS entry of intravenously delivered macromolecules was also induced by the FDA-approved, adenosine A2A receptor agonist Lexiscan: 10 kDa dextran was detectable within the CNS of mice as soon as 5 min after drug injection (Carman et al., 2011). Actually, adenosine receptor activation by agonists was indeed associated with augmented actinomyosin stress fiber formation indicating that adenosine receptors signaling initiates changes in cytoskeletal organization and cell shape. These processes are reversed as the half-life of the adenosine receptor agonist decreases. In agreement with these findings and with the observation that human brain endothelial cells do respond to adenosine in vitro, agonist-induced A2A receptor signaling transiently permeabilized a primary human brain endothelial cell monolayer to the passage of both drugs and human T cells in vitro (Kim and Bynoe, 2014).

Hence, by regulating the expression level of factors crucially involved in tight junction integrity/function, signaling induced through receptors for adenosine acts as a potent, endogenous modulator of BBB permeability in mouse models or human brain endothelial cells.

2. Adenosine receptor signaling increases immune cell entry by increasing BBB permeability.

Adenosine contributes to restraining leukocyte recruitment and platelet aggregation and might be important to control vascular inflammation. This is notable as most studies show that leukocyte migration into the CNS or in vitro BBB models occur by both paracellular and transcellular pathways (Zimmermann, 1992; Yegutkin, 2008). Thus, in vivo paracellular T-cell transendothelial migration under physiological conditions may be mediated by adenosine receptor signaling. Previous study observed that CD73-generated adenosine promotes the entry of inflammatory lymphocytes into the CNS during EAE development (Mills et al., 2008). For adenosine to exert biological effect, CD73 and adenosine receptors must be present on the same cell or on adjacent cells, because adenosine acts locally due to its short half-life. CD73, adenosine A1 and adenosine A2A receptor are indeed expressed on BBB endothelial cells in mice and humans. While CD73 is highly and constitutively expressed on choroid plexus epithelial cells that form the blood to CSF barrier, its expression on brain endothelial barrier cells is low under steady state conditions.

In addition, selective adenosine A2A receptor agonist CGS21680 caused an increase in CX3CL1 level in the brain of treated mice. Conversely, the A2A adenosine receptor antagonist SCH58261 protected mice from CNS lymphocyte infiltration and EAE induction recapitulating the phenotype of CD73 null mutant mice (Imai et al.,1997; Mills et al., 2012). Thus, the augmented CX3CL1 expression level seen in the brain of EAE developing mice can be regulated by adenosine A2A receptor signaling. This suggests that CD73/A2A receptor signaling may preferentially regulate inflammatory immune cells entry into the CNS but confers less stringency on these suppressor T cells.

Moreover, pharmacological activation or inhibition of the adenosine A2A receptor expressed on BBB cells opens and tightens the BBB, respectively, to entry of macromolecules or cells in the mice. The observation that adenosine can modulate BBB permeability upon A2A receptor activation suggest that this pathway might represent a valuable strategy for modulating BBB permeability and promote drug delivery within the CNS (Kim and Bynoe, 2014). Especially, FDA-approved, adenosine A2A receptor agonist, Lexiscan, or a broad-spectrum agonist, NECA, increased BBB permeability and supported macromolecule delivery to the CNS (Carman et al., 2011).

In other words, inhibiting adenosine receptor signaling on BBB cells restricts the entry of macromolecules and inflammatory immune cells into the CNS with limited impact on anti-inflammatory, T regulatory cells. Conversely, activation of adenosine receptors on BBB cells promotes entry of small molecules and macromolecules in the CNS in a time-dependent manner. The duration of BBB permeabilization depends on the half-life of the adenosine receptor activating agent or agonist, suggesting that adenosine receptor modulation of the BBB is a tunable system. 

These actions can eventually be seen as a predictable mechanism for the in vivo actions of PDRN in relation to BBB permeability through adenosine A2A receptor stimulation. PDRN acting on adenosine A2 subtype receptors in the cerebrovasculature (increase in E and I prostaglandin levels with consequent regional vasodilatation, inhibition of platelet aggregation, reduction of leukotriene B4 levels) may increase oxygen and substrate supply using a range of doses between 30 and 200 mg/kg in different species (Palmer and Goa, 1993). Furthermore, PDRN has been shown to directly inhibit lipid peroxidation activated during ischemic events in the rats. In addition, PDRN has been shown to inhibit the activation of neutrophils through adenosine A2 subtype receptors, so indirectly reducing the release of free oxygen radicals and other cytotoxic substances (Di Perri et al., 1991).

It is likely that an active carried-mediated transport enables PDRN to traverse the blood-brain barrier as already reported for nucleosides (Pardridge, 1983; Paschen et al., 1988). Finally, it can be hypothesized that PDRN could interact with polyamines whose synthesis can be greatly stimulated in response to ischemia.

<References>

Carman AJ, Mills JH, Krenz A, Kim DG, Bynoe MS. Adenosine receptor signaling modulates permeability of the blood-brain barrier. J Neurosci. 2011;31(37):13272-80. 

Di Perri T, Pasini FL, Ceccatelli L, Pasqui AL, Capecchi PL. Defibrotide inhibits Ca2+ dependent neutrophil activation: implications for its pharmacological activity in vascular disorders. Angiology. 1991;42(12):971-8. 

Imai T, Hieshima K, Haskell C, Baba M, Nagira M, Nishimura M, Kakizaki M, Takagi S, Nomiyama H, Schall TJ, Yoshie O. Identification and molecular characterization of fractalkine receptor CX3CR1, which mediates both leukocyte migration and adhesion. Cell. 1997;91(4):521-30. 

Kim DG, Bynoe MS. A2A Adenosine Receptor Regulates the Human Blood-Brain Barrier Permeability. Mol Neurobiol. 2015;52(1):664-78. 

Koszalka P, Ozüyaman B, Huo Y, Zernecke A, Flögel U, Braun N, Buchheiser A, Decking UK, Smith ML, Sévigny J, Gear A, Weber AA, Molojavyi A, Ding Z, Weber C, Ley K, Zimmermann H, Gödecke A, Schrader J. Targeted disruption of cd73/ecto-5'-nucleotidase alters thromboregulation and augments vascular inflammatory response. Circ Res. 2004;95(8):814-21. 

Mills JH, Alabanza LM, Mahamed DA, Bynoe MS. Extracellular adenosine signaling induces CX3CL1 expression in the brain to promote experimental autoimmune encephalomyelitis. J Neuroinflamm. 2012;9:193. 

Mills JH, Thompson LF, Mueller C, Waickman AT, Jalkanen S, Niemela J, Airas L, Bynoe MS. CD73 is required for efficient entry of lymphocytes into the central nervous system during experimental autoimmune encephalomyelitis. Proc Natl Acad Sci USA. 2008;105(27):9325-30. 

Palmer KJ, Goa KL. Defibrotide. A review of its pharmacodynamic and pharmacokinetic properties, and therapeutic use in vascular disorders. Drugs. 1993;45(2):259-94. 

Pardridge WM. Neuropeptides and the blood brain barrier. Annu Rev Physiol. 1983;45:73-82.

Paschen W, Schmidt-Kastner R, Hallmayer J, Djuricic B. Polyamines in cerebral ischemia. Neurochem Pathol. 1988;9:1-20.

Yegutkin GG. Nucleotide- and nucleoside-converting ectoenzymes:important modulators of purinergic signalling cascade. Biochim Biophys Acta. 2008;1783(5):673-94. 

Zimmermann H. 5′-Nucleotidase: molecular structure and functional aspects. Biochem J. 1992;285(2):345-65.

---

## [Decision Letter · Decision Letter 1]

12 Feb 2021

PONE-D-20-25445R1

Adenosine A2A receptor agonist polydeoxyribonucleotide ameliorates short-term memory impairment by suppressing cerebral ischemia-induced inflammation via MAPK pathway

PLOS ONE

Dear Dr. Han,

Thank you for submitting your manuscript to PLOS ONE. After careful consideration, we feel that it has merit but does not fully meet PLOS ONE’s publication criteria as it currently stands. Therefore, we invite you to submit a revised version of the manuscript that addresses the points raised during the review process.

We look forward to receiving your revised manuscript.

Kind regards,

Giuseppe Pignataro, MD, PhD

Academic Editor

PLOS ONE

Reviewers' comments:

Reviewer's Responses to Questions

**Comments to the Author**

1. If the authors have adequately addressed your comments raised in a previous round of review and you feel that this manuscript is now acceptable for publication, you may indicate that here to bypass the “Comments to the Author” section, enter your conflict of interest statement in the “Confidential to Editor” section, and submit your "Accept" recommendation.

Reviewer #1: All comments have been addressed

Reviewer #2: (No Response)

2. Is the manuscript technically sound, and do the data support the conclusions?

Reviewer #1: Yes

Reviewer #2: Yes

3. Has the statistical analysis been performed appropriately and rigorously? 

Reviewer #1: Yes

Reviewer #2: Yes

4. Have the authors made all data underlying the findings in their manuscript fully available?

Reviewer #1: Yes

Reviewer #2: Yes

5. Is the manuscript presented in an intelligible fashion and written in standard English?

Reviewer #1: Yes

Reviewer #2: Yes

6. Review Comments to the Author

Reviewer #1: The manuscript is in line with the previous comments.

The abstract lacks a reference to the brain areas where the measurements were made.

Reviewer #2: The present paper by Il-Gyo Ko and collegues aims to demonstrate the therapeutic effect of PDRN (polydeoxyribonucleotide), an adenosine A2A receptor agonist, on cerebral ischemia in gerbils by suppressing the secretion of pro-inflammatory cytokines with an anti-inflammatory effect. This action is accompanied by increased phosphorylation of MAPK in hippocampus and an amelioration in short-term memory in gerbils. The model of global cerebral ischemia performed is of 2-common carotid arteries occlusion for 7 minutes.

The manuscript is not original, since there is a lot of evidence about the neuroprotective role for PDRN in cerebral ischemia in rodents. At the same time, it has been widely demonstrated the involvement of A2A adenosine receptors in cerebral ischemia reperfusion injury, in the signaling to phosphorylated extracellular signal-regulated protein kinase (pERK1/2). The survival pathway evoked is clearly linked to an amelioration in short-term memory performances in rodent models of global cerebral ischemia, where CA1 ippocampal populations are mainly impaired. Therefore the paper is not innovative, not suitable of publication on PLOS journal.

7. PLOS authors have the option to publish the peer review history of their article (what does this mean?). If published, this will include your full peer review and any attached files.

Reviewer #1: **Yes: **Mario Campanile

Reviewer #2: No

---

## [Author Response · Author response to Decision Letter 1]

1 Mar 2021

Answers to Reviewers’ Comments

Manuscript ID: PONE-D-20-25445

Title: Adenosine A2A receptor agonist polydeoxyribonucleotide ameliorates short-term memory impairment by suppressing cerebral ischemia-induced inflammation via MAPK pathway

Authors: Il-Gyu Ko, Jun-Jang Jin, Lakkyong Hwang, Sang-Hoon Kim, Chang-Ju Kim, Jung Won Jeon, Jun-Young Chung, Jin Hee Han

We sincerely appreciate for your kind advice and comments to our manuscript. We revised the manuscript according to the reviewer’s comments. We added new experimental data, and modifications were expressed in red. 

Reviewer #1 

The manuscript is in line with the previous comments. The abstract lacks a reference to the brain areas where the measurements were made.

Answer: According to reviewer comment, we have added word in the abstract.

• Following word was added into the Abstract.

In the current study, induction of ischemia enhanced the levels of pro-inflammatory cytokines and increased phosphorylation of MAPK signaling factors in the hippocampus and basolateral amygdala.

Reviewer #2

The present paper by Il-Gyo Ko and collegues aims to demonstrate the therapeutic effect of PDRN (polydeoxyribonucleotide), an adenosine A2A receptor agonist, on cerebral ischemia in gerbils by suppressing the secretion of pro-inflammatory cytokines with an anti-inflammatory effect. This action is accompanied by increased phosphorylation of MAPK in hippocampus and an amelioration in short-term memory in gerbils. The model of global cerebral ischemia performed is of 2-common carotid arteries occlusion for 7 minutes.

The manuscript is not original, since there is a lot of evidence about the neuroprotective role for PDRN in cerebral ischemia in rodents. At the same time, it has been widely demonstrated the involvement of A2A adenosine receptors in cerebral ischemia reperfusion injury, in the signaling to phosphorylated extracellular signal-regulated protein kinase (pERK1/2). The survival pathway evoked is clearly linked to an amelioration in short-term memory performances in rodent models of global cerebral ischemia, where CA1 hippocampal populations are mainly impaired. Therefore, the paper is not innovative, not suitable of publication on PLOS journal.

Answer: A search for "PDRN" and "ischemia" in PubMed only shows 6 results: renal ischemia, skin ischemia, testicular ischemia, and peripheral arterial occlusion. There have been no studies on the effect of PDRN on the central nervous system, particularly the brain. Therefore, this study is considered to be the first to evaluate the action and mechanism of PDRN on brain ischemia. In this study, we first identified the pharmacological action of PDRN in brain ischemia, and proved that this paper has sufficient originality.

We have demonstrated that PDRN treatment promoted the signaling of the cAMP-PKA pathway by activating the adenosine A2A receptor and enhanced phosphorylation of CREB. It was also found that increased cAMP concentration reduced the expression of pro-inflammatory cytokines by inhibiting the phosphorylation level of the MAPK cascade pathway. This mechanism of action of PDRN has been proven to be effective in ameliorating short-term memory loss. In this paper, we identified the mechanism of action of PDRN during brain ischemia, and contained very innovative contents.

In addition, the PDRN used in this study is one of many adenosine A2A receptor agonists, whose origin differs from conventional drugs. PDRN (source: DNA extracted from salmon sperm cells) used in this study has very high stability and pharmacological action unlike compounds extracted from mammalian organs or PDRN (eg Defibrotide). Thus, this study evaluated the therapeutic efficacy of cerebral ischemia using PDRN, a highly stable drug. In this paper, we deal with PDRN, a very stable drug, and contain novel discoveries.

Therefore, we disagree with the reviewers' opinion that our work is not innovative. We think this paper has originality containing innovative contents and novel discoveries.

---

## [Editor Report · Decision Letter 2]

4 Mar 2021

Adenosine A2A receptor agonist polydeoxyribonucleotide ameliorates short-term memory impairment by suppressing cerebral ischemia-induced inflammation via MAPK pathway

PONE-D-20-25445R2

Dear Dr. Han,

We’re pleased to inform you that your manuscript has been judged scientifically suitable for publication and will be formally accepted for publication once it meets all outstanding technical requirements.

Kind regards,

Giuseppe Pignataro, MD, PhD

Academic Editor

PLOS ONE
---

## [Editor Report · Acceptance letter]

9 Mar 2021

PONE-D-20-25445R2 

Adenosine A_2A_ receptor agonist polydeoxyribonucleotide ameliorates short-term memory impairment by suppressing cerebral ischemia-induced inflammation via MAPK pathway 

Dear Dr. Han:

I'm pleased to inform you that your manuscript has been deemed suitable for publication in PLOS ONE. Congratulations! Your manuscript is now with our production department. 

Kind regards, 

on behalf of

Prof. Giuseppe Pignataro 

Academic Editor

PLOS ONE